# Synthesis of monodisperse InSb colloidal quantum dots by monomer concentration control for short-wave infrared photodetectors

Lucheng Peng [1], Miguel Dosil[1], Debranjan Mandal[1], Hao Wu[1], Aditya Malla [1] & Gerasimos Konstantatos [1,2] ✉

InSb colloidal quantum dots combine a low bulk bandgap (0.17 eV) with a large exciton Bohr radius, enabling access to short-wave infrared wavelength within the quantum confinement regime, alongside strong covalent bonding, complementary metal-oxide semiconductor compatibility, and restriction of hazardous substances compliance. However, prior one-pot and hot-injection approaches yield broad size distributions and weak excitonic absorption, while continuous-injection methods improve spectral features but are restricted to small dot sizes (<1.2 μm excitonic peaks). Here, we introduce a monomer-concentration-controlled approach that produces narrow-size-dispersed InSb quantum dots tunable from 950 to 1900 nm with the sharpest excitonic absorption peaks reported to date. Their monodisperse nature allowed the emergence of heavy hole-light hole splitting evident in their optical absorption spectra. Leveraging these high-quality nanocrystals, we demonstrate short-wave infrared photodetectors achieving external quantum efficiencies of 22% at 1500 nm and 19% at 1580 nm, extending the spectral reach of III-V colloidal quantum dot photodetectors at this wavelength range.

Environmentally friendly InSb colloidal quantum dots (CQDs) exhibit multiple advantages that include low bulk bandgap (0.17 eV) with large exciton Bohr radius (∼ 60 nm) allowing access to short-wave infrared wavelength within the quantum confinement regime[1], strong covalent bond character enabling solvent tolerance and thermal stability[2], and complementary metal-oxide semiconductor (CMOS) integrability and compliance with RoHS (restriction of hazardous substances) regulations. These attributes render InSb CQDs promising for a range of optoelectronic applications in the short-wave infrared window, such as LIDAR and 3D imaging for automotive and augmented/virtual reality (AR/VR)[3], night vision for surveillance[4], spectroscopy for food quality inspection[5,6], and bioimaging[7–9], provided that the high-quality InSb CQDs is accessible.

In comparison to the II-VI, IV-VI and III-V analogue CQDs, the synthesis of InSb CQDs remains underdeveloped despite recent advances[10–19]. To date, the synthetic strategies for InSb CQDs can be broadly categorized into three approaches: the one-pot heating up method[10,11,13,14,16,17], hot-injection method[12,16,18] and continuous-injection method[15,19]. However, CQDs prepared by the first two approaches typically exhibit broad size dispersity, as reflected in poorly defined absorption features even after size selection. In contrast, InSb CQDs synthesized by the continuous-injection method display more distinct exciton absorption peaks[15,19], yet the availability of monodisperse samples is still restricted to relatively small sizes (corresponding to excitonic peaks below 1.2 μm)[19]. The weak first excitonic absorption of InSb CQDs is often attributed to

[1]ICFO-Insitut de Ciencies Fotoniques, The Barcelona Institute of Science and Technology, Castelldefels, Barcelona, Spain. [2]ICREA-Institució Catalana de Recerca i Estudis Avançats, Lluis Companys 23, Barcelona, Spain. ✉e-mail: gerasimos.konstantatos@icfo.eu

(i) strong quantum confinement effects due to the large exciton Bohr radius, where small size variations broaden the absorption, and (ii) surface oxidation due to InSb sensitivity to water and oxygen[14,18]. Importantly, prior work[19] has highlighted the importance of having narrow-size-dispersed InSb CQDs for high-performance SWIR photodetectors.

To address the existing challenges in the continuous injection-based synthesis of InSb CQDs and enable high-quality InSb CQDs suitable for advanced optoelectronic applications, it is crucial to understand the mechanisms that govern CQD formation and growth. Currently, the growth mechanism for InSb CQDs remains unclear and highly dependent on the synthetic protocol. For instance, for the hot injection approach, Busatto et al. propose the aggregative coalescence growth mechanism of InSb CQDs[12], whereas Wang et al. propose the non-classical growth mechanism of InSb CQDs, which follows an initial formation of amorphous intermediate and a subsequent stepwise crystallization process[18]. Consequently, a better understanding of InSb CQD nucleation and growth is required to guide the development of improved synthetic strategies.

In this work, we shed light on the underlying mechanisms of InSb CQD growth based on the continuous injection approach and describe a synthetic route that overcomes the shortcomings of previous works. Growth kinetics investigation reveals that the synthesis of InSb CQDs based on the continuous injection follows the classic nucleation theory, and the resultant broad size distribution previously[15] originates from the presence of continuous nucleation events due to the persistent monomer supersaturation. In this work, we introduce the monomer concentration control approach to synthesize nearly-monodisperse luminescent InSb CQDs tunable from 950-1900 nm with the sharpest excitonic absorption peaks (and consequently narrow size distribution) reported to date. Based on this synthetic approach, we further report InSb CQD short-wave infrared (SWIR) photodetectors with the external quantum efficiency (EQE) of 22% at 1500 nm and 19% at 1580 nm which exceeds all reported heavy metal-free CQD-based SWIR photodiodes.

## Results and discussion

### Growth kinetics insights towards monodisperse InSb quantum dots

It is widely accepted that the size and size distribution of QDs can be well controlled by LaMer model based on classical nucleation theory[20]. This model describes the concept of burst nucleation driven by supersaturation of monomer and diffusion-controlled growth, but it ignores the prenucleation stage of the crystallization of QDs that could lead to the formation of persistent monomer supersaturation resulting in continuous nucleation events[20]. In continuous nucleation, the nucleation and growth processes are coupled with each other described by Avrami et al.[21,22], which is considered detrimental for achieving monodispersed colloidal QDs. The impact of continuous nucleation on the size and size distribution of II-VI and IV-VI QDs can be readily mitigated either by selecting appropriate precursors[23] or through homogenization/alloying due to the nature of the ionic bond in ionic QDs compounds[24,25]. In the case of III-V QDs - InP in particular, the continuous nucleation event has also been observed[26,27], and efforts to address it have been based on tuning the reactivity of the metal cation precursors[27]. In contrast, for the InSb CQDs, with the strongest covalent bond among all III-V semiconductors and with limited precursor options, the impact of continuous nucleation is particularly challenging to address.

To understand the growth kinetics of InSb CQDs, we monitored the growth process of InSb CQDs by absorption spectra based on the continuous injection protocol (as shown in Fig. 1a, b). By normalizing the absorption intensity (Fig. 1c) and fitting the experimental kinetics with the Avrami formula (Fig. 1d, details of fitting described in Methods), the Avrami exponent n is calculated to be 2.04, indicating the growth mechanism of InSb CQDs in previously reported continuous injection synthesis is dominated by continuous nucleation[20]. We hypothesized that the continuous nucleation is caused by persistent supersaturation of monomers due to the fast continuous injection rate of the precursor, resulting in broad size distribution as described in Fig. 1e. To tackle this problem, we developed the monomer-concentration-control approach (MCCA)

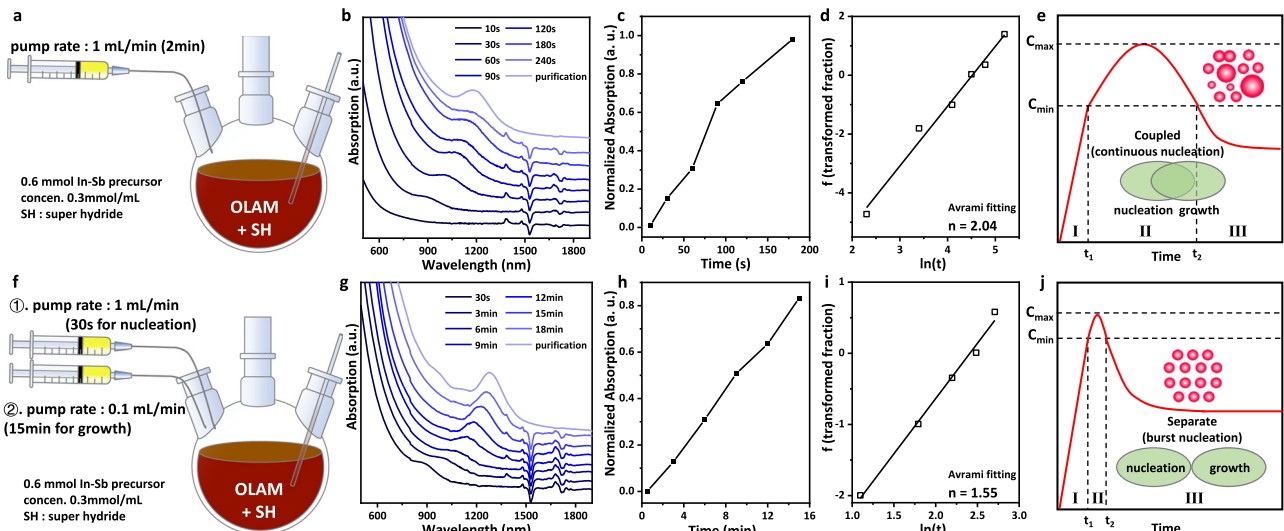

**Fig. 1 | Kinetic insights into the growth of InSb CQDs.** Schematic representation of (**a**) the previous synthesis method and (**f**) the monomer-concentration-control approach (MCCA). **b**–**d** the growth process monitor of InSb CQDs at 240 °C (**b**), normalization of the absorption intensity (**c**), and experimental kinetics fitting with Avrami formula (**d**) based on the previous synthesis method. **g**–**i** the growth process monitor of InSb CQDs at 240 °C (**g**), normalization of the absorption intensity (**h**), and experimental kinetics fitting with Avrami formula (**i**) based on the MCCA. Schematic representation of the temporal evolution of monomer concentration corresponding to the previous synthesis method (**e**) and the MCCA (**j**). It should be noted that the schematic representation in (**e**) and (**j**) are summarized according to the growth process and do not correspond to quantitative experimental data. The previous synthesis method exhibits a continuous nucleation growth mode, while the MCCA indicates a burst nucleation growth mode.

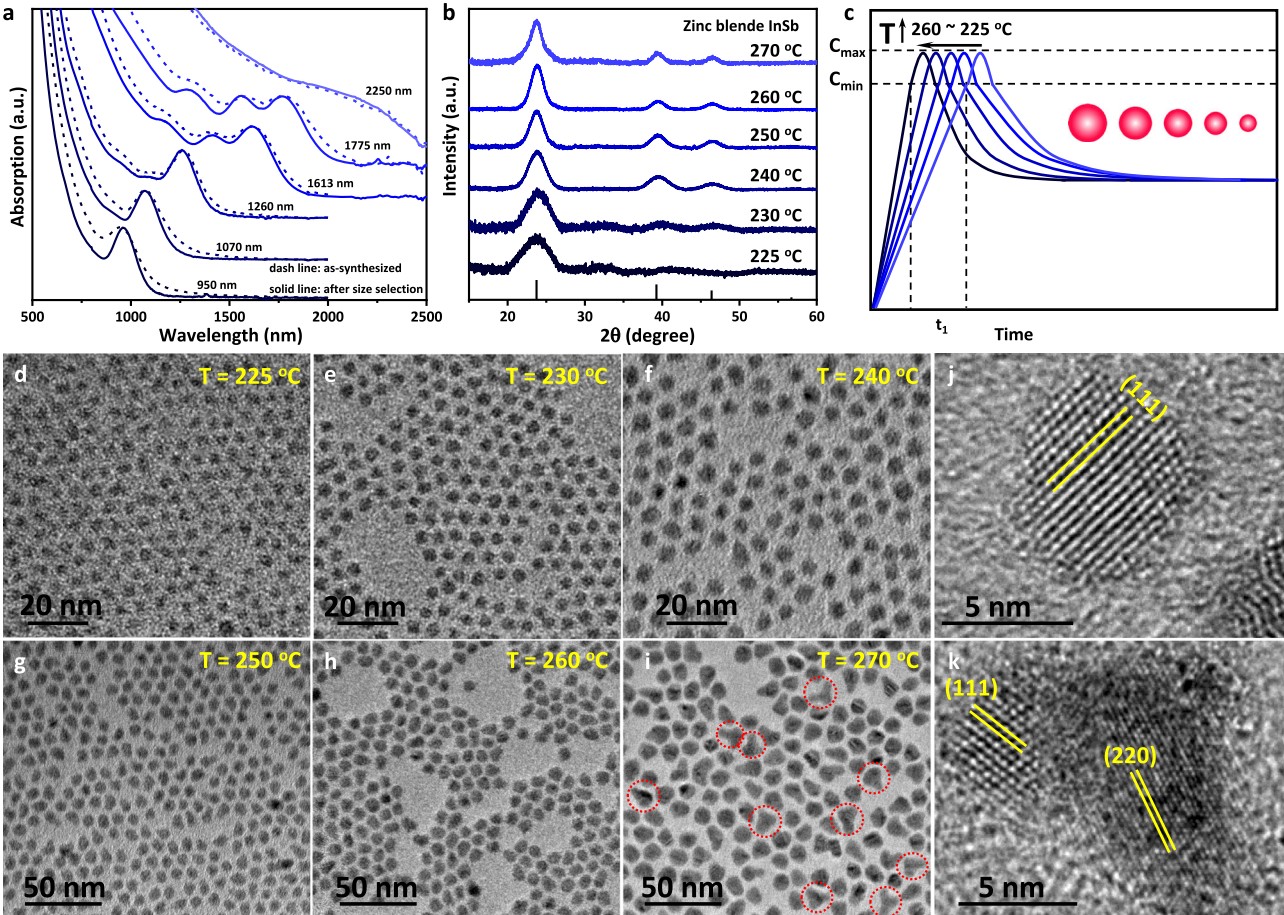

**Fig. 2 | Effect of reaction temperature on the size of InSb CQDs. a** Absorption spectra of InSb CQDs obtained at 225 °C, 230 °C, 240 °C, 250 °C, 260 °C, and 270 °C respectively. **b** Powder X-ray diffraction (XRD) patterns of InSb CQDs corresponding to (**a**), identical to the zinc blende crystal structure. **c** Schematic representation of the temporal evolution of monomer concentration with variation of reaction temperature. It should be noted that this does not correspond to quantitative experimental data. **d**–**i** Transmission electron microscope (TEM) of InSb CQDs obtained at 225 °C, 230 °C, 240 °C, 250 °C, 260 °C, and 270 °C with the size corresponding to 2.5, 3.2, 4.3, 6.2, 7.1, and 10.5 nm. **j** The high-resolution TEM image of sphere-shaped InSb CQD obtained at 250 °C. **k** The high-resolution TEM image of quasi-tetrapod-shaped InSb CQD obtained at 270 °C due to the fusion.

by employing a fast injection rate of precursor for nucleation followed by a slow injection rate for growth, as demonstrated in Fig. 1f. The growth behavior is then monitored by taking the aliquots without any size-selection process for absorption measurement (Fig. 1g) to understand the growth kinetics. By normalizing the absorption intensity (Fig. 1h) and fitting the experimental kinetics with Avrami formula (Fig. 1i), the Avrami exponent n is calculated to be 1.55 that matches with the burst nucleation model[20]. Interestingly, comparing Fig. 1b with 1g, we observe a strong narrowing of the size distribution, here reflected by the peak-to-valley ratio of the first excitonic peak. Figure 1j demonstrated the formation and growth process of InSb CQDs based on the MCCA, and the monodisperse InSb CQDs is obtained by controlling the monomer concentration to avoid continuous nucleation. To probe the extent to which the precursor addition rate affects the quality of InSb CQDs, we varied the pumping speed for growth from 1 mL/min to 0.2 mL/min to 0.1 mL/min to 0.05 mL/min. Supplementary Fig. 1 shows that the QD size distribution broadens upon increasing the addition rate, evident from the band-tail absorption and the decreasing peak-to-valley ratio of the excitonic peak. Notably, the faster the addition rate, the more the absorption spectrum blue-shifts. Considering that the total amount of precursor is fixed, a faster addition rate induces continuous nucleation, thereby consuming the available precursors for growth and resulting in smaller QDs.

## Experimental parameter optimization for size-tunable InSb quantum dots

We attempted to optimize the experimental parameters for monodisperse size-tunable InSb CQDs according to the above growth model analysis. First, we sought to investigate the effect of the reaction temperature. As shown in Fig. 2a, the absorption spectra of InSb CQDs exhibit sharp excitonic peaks, especially after size selection, whose positions were controlled from 950 to 1775 nm by varying the reaction temperatures from 225 to 260 °C. Notably, the InSb CQDs exhibit better excitonic absorption peaks even before size selection compared to previous reports[11,12,16]. Transmission electron microscopy (TEM) images (Fig. 2d–i) confirm the InSb CQDs with average sizes of 2.5, 3.2, 4.3, 6.2, 7.1, and 10.5 nm that correspond to the temperatures at 225 °C, 230 °C, 240 °C, 250 °C, 260 °C, and 270 °C, respectively. More TEM images on a large scale are shown in Supplementary Figs. 2–7. We notice that when the temperature increases to 270 °C, the InSb CQDs redshift to 2250 nm, but are also accompanied by a weakening of the excitonic absorption peak. We attribute this deterioration to morphology evolution, as the formation of quasi-tetrapod shaped QDs is observed in TEM images which is likely caused by the partial fusion of QDs at high temperature (Fig. 2i, k, and S7). The powder X-ray diffraction patterns (XRD) shown in Fig. 2b confirm a pure cubic zinc blende crystal structure of InSb CQDs, identical to that of bulk InSb. Figure 2c summarizes the effect of temperature on the size of InSb

CQDs. We posit that as the temperature increases, the conversion rate of precursors to monomers accelerates[28], shortening the time required to reach monomer supersaturation for nucleation. Consequently, more precursors are consumed during the growth stage, resulting in larger InSb CQDs. To test this hypothesis, we first determined the molar absorption coefficient ($\varepsilon_{E1}$) of InSb CQDs at the first exciton transition energy that can be fit by the expression $\varepsilon_{E1} = 12103d^{1.756}$ (Supplementary Fig. 8). Noteworthily, the extinction coefficient scales with dot diameter as $\varepsilon \propto d^{1.8}$, which deviates from the theoretical prediction ($\varepsilon \propto d^3$), yet is comparable to that reported for other III-V InAs CQDs ($\varepsilon \propto d^{1.28}$)[29]. This behavior indicates non-volumetric absorption, probably due to the extreme quantum confinement, the large exciton Bohr radius, surface contributions in the electronic structure of InSb CQDs, and merits further investigation. The absorption spectra of InSb CQDs synthesized at 230 °C, 240 °C, and 250 °C using the same precursor amount are compared in Supplementary Fig. 9. Based on Beer-Lambert law calculation, it is found that the InSb CQDs obtained at higher temperatures exhibit lower QD concentration, indicating fewer nuclei formed, which is consistent with our conclusion.

Furthermore, considering the observation of fusion behavior between QDs due to the accelerated ligand exchange dynamics under high-temperature conditions[30,31], we sought to tune the size of InSb CQDs by increasing the amount of precursor at relatively lower temperatures. This motivation arises from the fact that oleylamine is a relatively weak ligand for InSb CQDs, necessitating the addition of oleic acid during purification to prevent QD aggregation[10]. Moreover, QD fusion has been shown to occur during the synthesis, as evidenced by the band-tail absorption features and the corresponding TEM images reported in our previous work[19]. Figure 3a–c show the absorption spectra of InSb CQDs synthesized at 240-260 °C when the precursor amount is increased from 0.6 mmol to 0.9 mmol to 1.2 mmol. At 240 °C, the absorption spectra redshift from 1250 nm to 1420 nm to 1570 nm; and at 250 °C, they redshift from 1610 nm to 1720 nm to 1900 nm. This monomer-addition-driven growth mode is also in strong agreement with the growth model that we analyzed above. Figure 3d–k presents the TEM images and the corresponding size distribution histograms are exhibited in Supplementary Fig. 10. The high quality InSb CQDs obtained by the reported method feature low size polydispersity of ~10%. More TEM images with different resolutions are shown in Supplementary Figs. 11-12. Notably, with high precursor concentration, especially at 260 °C, we observe absorption spectra broadening and the appearance of absorption tails of InSb CQDs. As with the high-temperature result, this is again caused by QD fusion. Hence, it is better to keep the monomer concentration below 50 µM for monodisperse InSb CQDs according to our study.

Figure 3l demonstrates the growth mechanism of monodisperse InSb CQDs based on the MCCA. The results discussed above indicate that it is crucial to control the monomer concentration during the growth process, as it can effectively suppress continuous nucleation, decouple nucleation from growth, and thereby narrow the size distribution. It should be noted that under high temperature (≥270 °C) and high QD concentration, the fast ligand adsorption-desorption equilibrium leads to the onset of fusion between InSb CQDs, resulting in a broader size distribution.

## Optical properties of InSb quantum dots

Having achieved the monodisperse InSb CQD synthesis, we next sought to explore their optical properties. The absorption spectra of representative InSb CQDs with band gap ranging from 1.3 eV to 0.65 eV are shown in Fig. 4a. The absorption spectra show the sharpest first absorption peaks among all III-V CQDs with the peak-to-valley ratio (1.8 ± 0.2). With the sharp excitonic absorption feature of InSb CQDs obtained above, size dependence of the band gap ($E_g$) of InSb CQDs is plotted in Fig. 4b. We first attempted to do the fitting by the theoretical model based on an eight-band effective-mass approximation (EMA)

approach, proposed by Efros and Rosen[32], which simultaneously takes into account the nonparabolicity of the electron- and light-hole dispersion and the coupling between conduction and valence bands. However, the fitting is unsuccessful and shows significant deviation that is in consistence with the report of Busatto et al.[12]. Subsequently, we turned to the tight-binding calculations model proposed by Allan and Delerue for fitting[33], which has been demonstrated well suited for InSb CQDs[12]. The tight-binding calculation describes as follow:

$$Eg(d) = Eg(\infty) + \frac{1}{ad^2 + bd + c} \quad (1)$$

where $E_g(d)$ and $E_g(\infty)$ are the band gap (in eV) for a QD of diameter d (in nm) and for bulk, respectively, and a, b, and c are constants. As shown in Supplementary Fig. 13, Eq. 1 has been successfully used to fit the experimentally observed size-dependent trends for InSb CQDs (Eq. 2).

$$Eg(d) = 0.17 + \frac{1}{0.0065d^2 + 0.1615d + 0.422} \quad (2)$$

However, as an empirical size-dependent equation, the a value of 0.0065 of $d^2$ is rather small, we therefore – for simplicity- fitted the $E_g$(d) solely as a function of $1/d$.

As shown in Eq. 3 (Fig. 4b), the empirical size-dependent band gap of InSb CQDs can be described as below:

$$Eg(d) = 0.17 + \frac{1}{0.2212d + 0.3032} \quad (3)$$

Moreover, the realization of narrow-size-dispersed InSb CQDs gives rise to the first demonstration of heavy-hole, light-hole (HH/LH) splitting reflected in the optical absorption spectrum of InSb CQDs. Due to the difference in the effective mass of heavy holes and light holes in III-V semiconductors, upon confinement, the LH states are split from HH towards higher energies, so that the valence band (VB) maximum is determined by the HH state and the LH state leads to a secondary absorption peak shifted towards higher energies (Fig. 4c)[34,35]. We found the split in energy ($\Delta E_{HH-LH}$) to scale linearly with confinement (Fig. 4d), as expected. The absorption and corresponding photoluminescence (PL) spectra of representative InSb CQDs with band gap ranging from 1.21 eV to 0.843 eV are presented in Fig. 4e, f. The PL is characterized by a Stokes shift of 20 to 90 meV depending on the size (Fig. 4e), and the full-width at half-maximum (FWHM) of the PL peaks is as narrow as 100 ± 10 meV (Fig. 4f), much narrower than previously reported works (130 meV − 150 meV)[12-16]. The Stokes shift behavior of InSb CQDs (i.e. the decreasing Stokes shift with size) may be likely due to a singlet-triplet splitting of the exciton states by the electron-hole exchange interaction instead of difference in energy between the exciton states formed with S and P hole states, as reported by Bagga et al for InAs QDs[36]. However, we also believe that this observation merits further theoretical and experimental investigations to pinpoint the origin of this effect for the case of InSb CQDs in the future. The size-dependent ($E_g$) light-heavy hole splitting values, PL FWHM, and Stokes shifts are summarized and presented in Supplementary Table 1. The PL quantum yield (PLQY) of InSb CQDs has been measured by the integration sphere method[15], with values ranging from 1 ~ 2%. The transient PL decay shown in Supplementary Fig. 14 can be fitted with a fast component, characteristic of a nonradiative recombination channel, followed by a slower nearly monoexponential component with average decay time of 70 ns corresponding to the radiative lifetime.

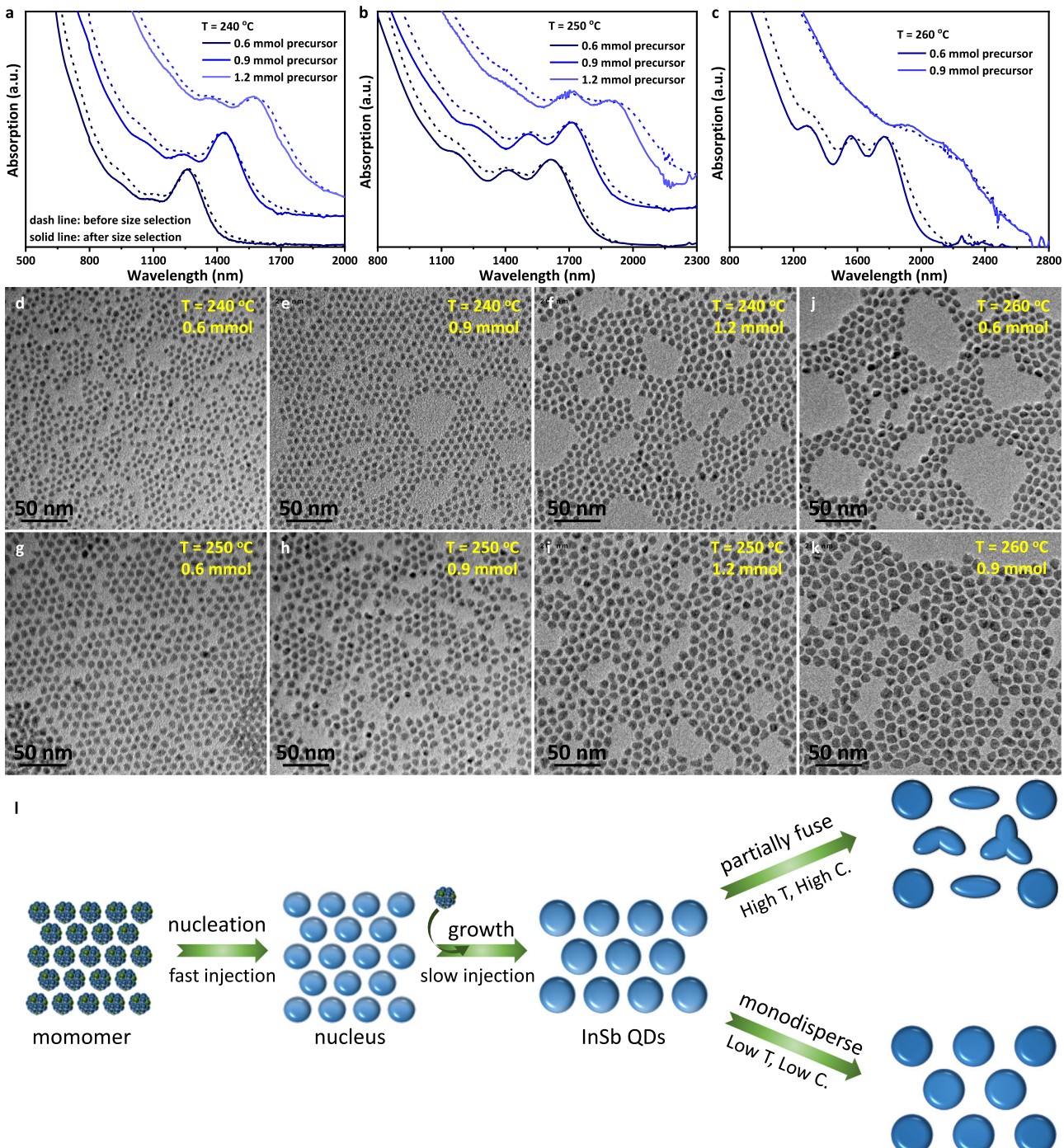

**Fig. 3 | Growth mechanism of InSb CQDs by monomer-concentration-control approach (MCCA). a–c** Absorption spectra of InSb CQDs obtained at 240 °C, 250 °C, 260 °C with the amount of precursor increased from 0.6 mmol to 1.2 mmol (**a**), 0.6 mmol to 1.2 mmol (**b**) and 0.6 mmol to 0.9 mmol (**c**) respectively. **d–k** TEM images of InSb CQDs corresponding to the absorption spectra in (**a–c**). **l** The growth mechanism of monodisperse InSb CQDs based on MCCA. The mechanism indicates that controlling the monomer concentration is crucial to suppress continuous nucleation during growth, thereby narrowing the size distribution. It should be noted that the high temperature and high QDs concentration can lead to the fusion of InSb CQDs, resulting in a broader size distribution.

## InSb/InP CQDs for short-wave infrared photodetectors

The operation wavelength of previous reported InSb CQDs SWIR photodetectors are generally below 1.45 μm[13,15,17,19,37] due to the challenges in synthesizing InSb CQDs. Thanks to the improvement of synthesis here, which makes the high-quality InSb CQDs accessible, hence provides the opportunity to fabricate a photodetector in which the wavelength is competitive with commercial Ge and InGaAs detectors (1.6 μm cut-off). We first employ the InSb CQDs with excitonic peak of 1465 nm to fabricate SWIR photodetectors with the device architecture of indium-tin-oxide (ITO)/TiO₂/InSb/PTAA/MoOx/Au as previously reported[15,19]. As shown in Supplementary Fig. 15, the device performance was very poor, characterized by high dark current under bias and low EQE, due to oxidation of InSb core CQDs as per our previous results[19]. Therefore, we decided to employ the InSb/InP core-shell structure in order to passivate the surface defects and suppress the formation of Sb-O species[15].

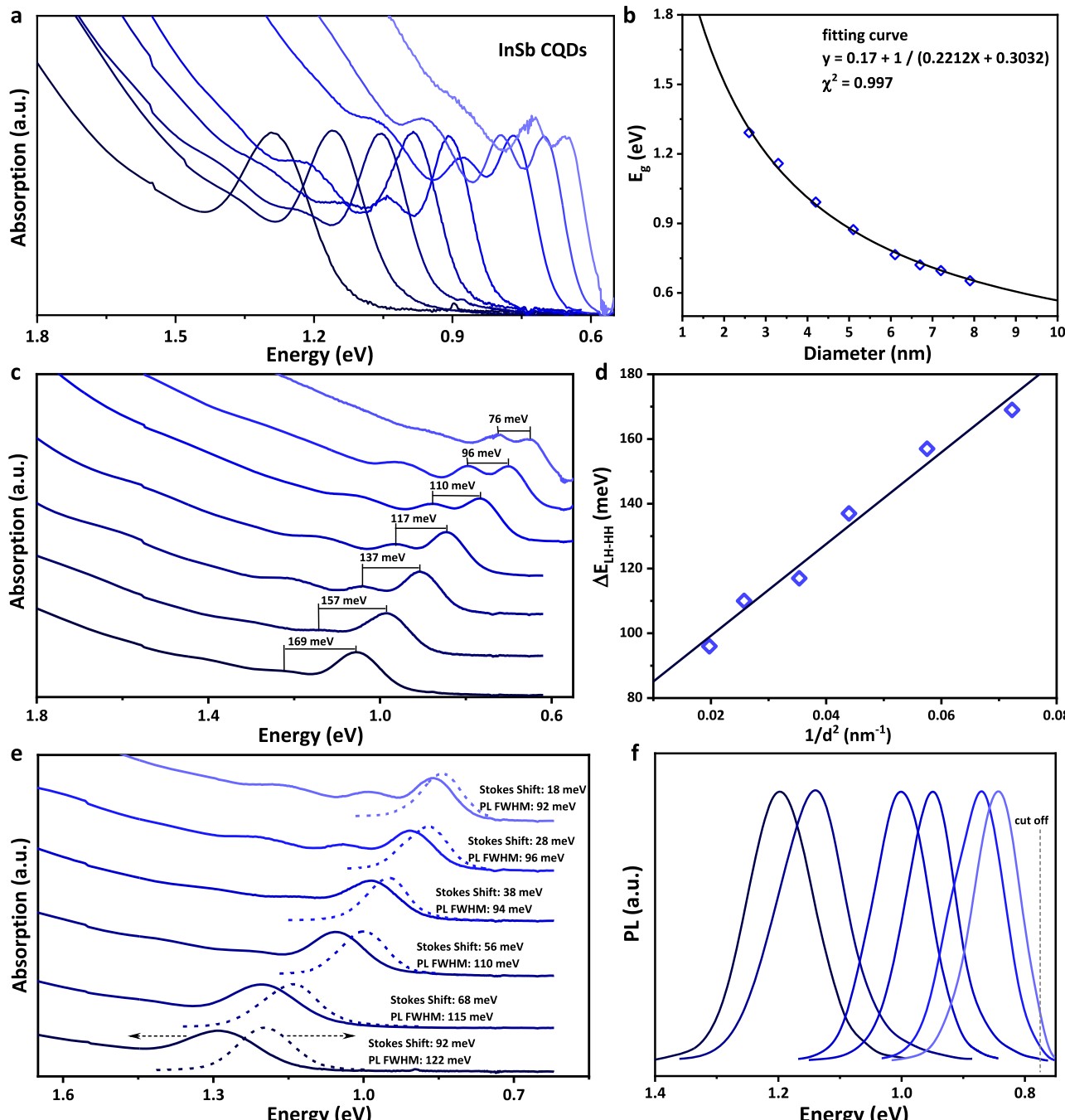

**Fig. 4 | Optical properties of monodisperse InSb CQDs. a** Absorption spectra of representative InSb CQDs with band gap ranging from 1.3 eV to 0.65 eV. The absorption spectra show the sharpest first absorption peaks among all InSb CQDs. **b** Size dependence of the band gap ($E_g$) of InSb CQDs obtained by MCCA and the fitting with tight-binding calculations by plotting the Eg ($d$) as a function of $1/d$. **c, d** Absorption spectra showing the valence band splitting energies (HH and LH) as function of CQD size. **e, f** The absorption and photoluminescence spectra of InSb CQDs with band gap ranging from 1.21 eV to 0.843 eV. The PL is clearly due to band-edge recombination since the Stokes shift is small (20 to 90 meV depending on the size). The full-width at half-maximum (FWHM) of the PL peaks is as narrow as $100 \pm 10$ meV.

The InSb CQDs with excitonic peak of 1360 nm is used as the core material. Upon InP shell growth, the exciton peak redshifts to 1475 nm (Fig. 5a). This behavior is consistent with our previous observation indicating the successful growth of core-shell structure. The corresponding TEM, EDX mapping, XRD, PL and PL lifetime characterizations are provided in Supplementary Fig. 16. The InSb/InP core-shell CQDs are then used to fabricate photodetectors with the architecture mentioned above. As shown in Fig. 5b, c, the rectification of current density-voltage (J-V) curve and external quantum efficiency (EQE) of the InSb/InP core-shell CQDs photodetector have shown significant

improvement than the core only photodetector, specifically with the dark current density of 45 µA cm$^{-2}$ at −1 V reverse bias and EQE of 8.5% at 1500 nm under 0 V bias, increasing to 22% under a reverse bias of −1 V. The responsivity (R), frequency-dependent current noise spectrum ($i_n$) and Specific detectivity spectrum (D* ~ $1.3 \times 10^{11}$ Jones @ 1 Hz) of this device are plotted in Supplementary Fig. 17. Additionally, the same strategy is applied to larger InSb CQDs, with the excitonic peak of 1440 nm for core and 1570 nm for core-shell (Fig. 5d), and the corresponding TEM, EDX mapping, XRD, PL and PL lifetime characterizations are provided in Supplementary Fig. 18. The resulting

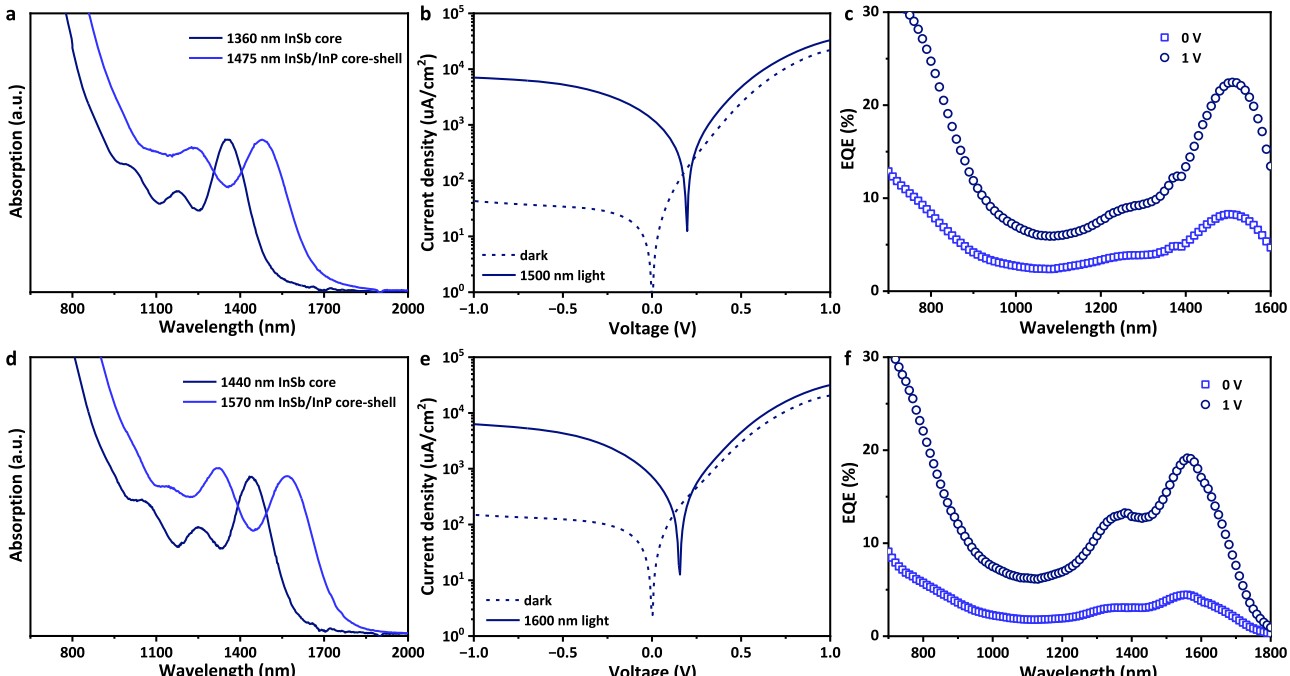

**Fig. 5 | InSb/InP CQDs short-wave infrared photodetector. a** Absorption spectra of InSb CQDs before and after InP shell growth. The first exciton absorption peak of InSb CQDs redshifts from 1360 nm to 1475 nm after InP shell growth. **b** Current density-voltage (*J-V*) curves of InSb/InP CQDs photodetector dark and under 1500 nm illumination with power densities of 47 mW cm⁻². **c** External quantum efficiency (EQE) spectra of InSb/InP CQDs photodetector with the bias of 0 V and 1 V (reverse bias). **d** Absorption spectra of InSb CQDs before and after InP shell growth. The first exciton absorption peak of InSb CQDs redshifts from 1440 nm to 1570 nm after InP shell growth. **e** Current density-voltage (*J-V*) curves of InSb/InP CQDs photodetector dark and under 1600 nm illumination with power densities of 27 mW cm⁻². **f** External quantum efficiency (EQE) spectra of InSb/InP CQDs photodetector with the bias of 0 V and 1 V (reverse bias).

photodetector achieves a dark current density of 150 μA cm⁻² at −1 V reverse bias, along with EQE of 4.5% (0 V) and 19% (−1 V) at 1580 nm (Fig. 5e, f). The responsivity (R), frequency-dependent current noise spectrum (*i*ₙ), and Specific detectivity spectrum (D* ∼ 3.3 × 10¹⁰ Jones @ 1 Hz) of this device are also provided in Supplementary Fig. 19. To be best of our knowledge, the EQE of InSb/InP CQDs photodetectors reported herein is the highest among all III-V photodetectors at the wavelength beyond 1500 nm. The conduction band, valence band, and Fermi level of the InSb/InP CQD films are measured by ultraviolet photoelectron spectroscopy (UPS, Supplementary Figs. 20, 21). We noticed that both films show similar intrinsic doping, and further device optimization developing p-type doped CQD layer as HTL would be needed to further suppress the dark current density under bias.

In summary, we developed a synthetic approach to produce monodisperse luminescent InSb CQDs with well-defined first excitonic peaks spanning 950-1900 nm and exhibiting high peak-to-valley ratios. The growth kinetics investigation reveals that precise control of the monomer concentration is essential to suppress continuous nucleation and to ensure that the growth of InSb CQDs proceeds according to classical nucleation theory, thereby narrowing both the size distribution and photoluminescence (PL) spectral linewidth. Leveraging these high quality InSb CQDs, we further fabricated SWIR photodetectors, which achieved the EQE of 22% at 1500 nm and 19% at 1580 nm, surpassing all previously reported heavy-metal-free CQD-based SWIR photodetectors. These results not only highlight the potential of InSb CQD-based SWIR photodetectors to access the wavelength range of Ge and InGaAs technology, but they also offer a promising path towards high-performance optoelectronic applications of InSb CQDs in the full SWIR. Last but not least, the realization of highly monodisperse InSb CQDs, as evidenced by the optical signature of the LH-HH splitting, offers a material platform suitable to investigate deeper into the photophysics of this class of CQDs.

## Methods

### Chemicals

Oleylamine (OLA, 80-90%), tris(trimethylsily)phosphine ((TMS)₃P, 98%) and octane (anhydrous, 99 + %) were purchased from Thermal Scientific. Indium(III) chloride (InCl₃, 99.999%), tris(dimethylamido) antimony-(III) (Sb[NMe₂]₃, 99.99%), and were purchased from Strem Chemicals. Indium acetate (In(OAc)₃, 99.99%), indium(III) iodide (InI₃, 99.998%), 1-octadecene (ODE, 90%), lithium triethylborohydride (LiEt₃BH, Super-Hydride, 1 M in tetrahydrofuran, THF), dioctyl ether (DOE, 99%), oleic acid (OA, 90%), methanol (anhydrous, 99.8%), toluene (anhydrous, 99.8%), Tris(pentafluorophenyl)borane (BCF, 95%) and methyl acetate (anhydrous, 99.5%) were purchased from Sigma Aldrich. Acetone (anhydrous, 98%) was purchased from Scharlab. Poly-[bis(4-phenyl)(2,4,6-trimethylphenyl)]amine (PTAA) was purchased from Ossila. All chemicals were used as received except for oleylamine and oleic acid, which were degassed before use. The degassing was performed at 100 °C under reduced pressure (∼1 mbar) for 4 h.

**Preparation of superhydride solution.** LiEt₃BH (1.0 M) dissolved in tetrahydrofuran (THF) is used as purchased. A 100 mL amount of the superhydride solution is added to 50 mL of degassed dioctyl ether (DOE) and then evacuated on the Schlenk line for >4 h, until the THF is completely removed. *Caution: LiEt₃BH is very reactive in air and should be handled using air-free processes. The final superhydride solution has a concentration of 2.0 M in DOE.*

**Preparation of the In-Sb Precursor Solution.** 1.6 mL toluene, 400 uL of OLA (1.2 mmol), and 115 uL of Sb[NMe₂]₃ were mixed together in a glovebox under N₂ (H₂O and O₂ < 0.1 ppm) yielding a bright yellow solution. Then 0.6 mmol InCl₃ was added and dissolved under stirring at room temperature for 5 min.

**Preparation of 0.2 M (TMS)₃P-ODE solution.** 1 mL (TMS)$_3$P solution was mixed with 16 mL degassed ODE, and the 0.2 M (TMS)$_3$P-ODE solution was formed.

**Synthesis of InSb colloidal quantum dots.** In a typical synthesis, 20 mL of degassed OLA was heated to 240 °C in a round-bottom flask under constant stirring and N$_2$. At this temperature, 4.5 mL of LiEt$_3$BH in DOE (2.0 M, 9 mmol) was added in a dropwise manner within 2 min. During the addition, OLA acquired an orange color. Subsequently, 2 mL of the In-Sb precursor solution was pumped into the organic mixture with the pumping rate of 1 mL min$^{-1}$ for 30 s for nucleation and 0.1 mL min$^{-1}$ for 15 mins for growth. After that, the temperature was kept at 240 °C for another 5 min and cooled down to room temperature naturally.

**InCl₃ Post-treatment for InSb colloidal quantum dots.** Before the purification of InSb CQDs, 1.2 mmol InCl$_3$ was loaded in a round-bottom flask filled with N2, and then the as synthesized InSb CQDs crude solution was transferred into the flask and heating to 200 °C from room temperature for 30 min for the post-treatment.

**Purification and Size-Selection of InSb Colloidal Quantum Dots.** After the post-treatment, the reaction solution was transferred into the plastic centrifuge tubes inside the glovebox. The reaction mixture was separated in two portions, oleic acid (2 mL) was added to each and rotated for 2 min. Then 24 mL of MeOH was added, leading to turbidity. Centrifugation for 10 min at 6000 rpm yielded a black precipitate and yellowish supernatant. The yellowish supernatant was discarded, and the black precipitate was redispersed in 10mL of toluene, and centrifuged for another 10 min at 6000rpm. The precipitate was discarded, then 2 mL oleic acid was added to the supernatant, and 15 mL of MeOH was added. Again, centrifugation for 10 min at 6000 rpm yielded a black precipitate and colorless supernatant. The colorless supernatant was discarded, and the black precipitate was redispersed in 4 mL anhydrous toluene. For the size-selection process, 4 mL of acetone was added into the quantum dots solution in toluene, and centrifugation for 10 min at 6000 rpm. The precipitate was discarded and the supernatant was added by another 20 mL of acetone and centrifugation for 10 min at 6000 rpm. The precipitate was collected and redispersed in 2 mL anhydrous toluene with a concentration of 50 mg mL$^{-1}$, yielding ~100 mg InSb CQDs per synthesis. The reaction yields of different CQD sizes are summarized in Supplementary Table 2.

**Synthesis of InSb/InP Core Shell Colloidal Quantum Dots.** In a typical synthesis, 0.1 mmol In(OAc)$_3$, 0.3 mmol OA, and 10 mL octadecene were mixed in a 25 mL flask, and evacuated on the Schlenk line at 100 °C for 2 h. Then the temperature was raised to 150 °C for the complete dissolution of In(OAc)$_3$. Next, the ODE solution was cooled down to 60 °C and 1 mL 50 mg mL$^{-1}$ InSb CQDs was added. After the addition of InSb CQDs, the solution was bubbled with N$_2$ for 30 min at 60 °C. Then 400 μL (TMS)3P-ODE solution (0.2M) was added, and the reaction temperature was raised to 270 °C for the growth of the InP shell. After 30 min growth of InP shell, the reaction mixture was cooled down to room temperature naturally.

**Purification of InSb/InP core shell colloidal quantum dots.** The reaction mixture was transferred into the plastic centrifuge tubes inside the glovebox and centrifuged for 10 min at 6000 rpm. The precipitate was discarded and 30 mL of methyl acetate was added to the supernatant. Centrifugation for 10 min at 6000 rpm yielded a black precipitate and colorless supernatant. The colorless supernatant was discarded, the black precipitate was redispersed in 10 mL of toluene, and the wash process was repeated. The final black precipitate was redispersed in anhydrous octane, resulting in a 50 mg mL$^{-1}$ InSb/InP

CQDs solution for further characterization and device fabrication. All the synthesis and purification steps were carried out in an inert atmosphere under anhydrous conditions (N$_2$ glovebox, H$_2$O and O$_2$ < 0.1 ppm).

**Avrami model fitting.** The growth kinetics of coupled nucleation and growth can be described by the Avrami model, which considers the influence of growth on the nucleation process. The general formula is expressed as:

$$f = 1 - exp(-Kt^n) \qquad (4)$$

where K is a constant representing the overall crystallization rate, including both nucleation rate and growth rate, $f$ is solution-to-solid transition volume fraction (here define as the normalized absorption intensity by integrating the absorption spectra) and n is the Avrami exponent.

By applying a logarithmic transformation, Eq. 4 can be converted into the following form:

$$ln(-ln(1-f)) = lnK + nlnt \qquad (5)$$

Figure 1d and i are plotted according to Eq. 5, where $f$ is the normalized absorption intensity and $t$ is the time in Fig. 1c and h, and the slope n yields the Avrami exponent.

**Characterization of the quantum dots.** UV-vis absorption measurements were performed with a Cary 5000 UV-Vis-NIR spectroscope in solution with N$_2$ filled atmosphere. For photoluminescence (PL) measurements, four channel Thorlabs laser was used as excitation light and a Kymera 328i spectrograph (Oxford Instruments, Andor) was used as the detector (<1600 nm). Steady-state photoluminescence (PL) of the solutions was measured by exciting diluted CQD dispersions in cuvettes with a continuous-wave 827 nm (1.50 eV) laser diode (LDH D-C-830, PicoQuant). The emitted PL was filtered using a long-pass filter (FELH850, Thorlabs) to suppress residual excitation and then focused into the entrance slit of a 328 mm spectrograph (Kymera 328i, Andor, Oxford Instruments) coupled to a 1D InGaAs array detector (iDus 1.7, Andor, Oxford Instruments). For transient PL measurements, the same laser diode operated in pulsed mode (70ps FWHM) was employed. The emitted light was filtered and fiber-coupled to a superconducting nanowire single-photon detector with a Gaussian instrument response function (IRF) of 24 ps FWHM (ID281, ID Quantique). Photon arrival times were recorded using a time-correlated single-photon counting module (ID1000, ID Quantique). For the PLQY measurement, a 935 nm laser, an integration sphere and the Andor detector were used. The XRD data were collected using a Rigaku SmartLab powder diffractometer in the Bragg–Brentano geometry with Cu Kα radiation on drop-casted powder samples. TEM was performed at the Scientific and Technological Centres of the University of Barcelona. The TEM images were obtained using a JEOL 2100 F microscope operating at an accelerating voltage of 200 kV. TEM samples were prepared by dropping CQD solution on ultrathin carbon grids.

**Inductively coupled plasma optical emission spectroscopy (ICP-OES).** ICP-OES measurements were performed on a PerkinElmer Optima 3200RL ICP-OES spectrometer at the Scientific and Technological Centers of the University of Barcelona. Samples were carefully dried under vacuum overnight and then thoroughly dissolved in HNO$_3$ (69.5%), yielding clean light yellow solutions. This solution was further diluted with HNO$_3$ 1% to reach the ppm ranges required for the measurement. In the meantime, In and Sb calibration curves (4 points) are prepared by diluting monoelemental 1 g/L certified standard solutions (Inorganic Ventures) with HNO3 1% to concentrations ranging from 0 to 10 ppm. All the calibration curves were fitted to lines with R-values

larger than 0.99. The molar absorption coefficient calculation and fitting of InSb CQDs is based on the ICP analysis of Sb element according to previously reported procedures[38].

**Device fabrication.** ITO-covered glass substrates (Universität Stuttgart, Institut für Großflächige Mikroelektronik) were cleaned by sonication in soapy water, acetone, and isopropanol for 20 min each and dried with nitrogen, followed by 30 min ultraviolet/ozone treatment. A TiO$_2$ electron transport layer was then deposited by sputtering to a thickness of ~40 nm. Three layers of InSb/InP CQDs were further spin-coated from 50 mg ml$^{-1}$ octane solution via the layer-by-layer method inside the glovebox. For each InSb/InP CQDs layer, one drop of QDs solution was spin-coated onto TiO$_2$/ITO substrates during spinning (2000 r.p.m.). Then, 5 mg ml$^{-1}$ InI$_3$/Methanol solution was applied to the CQDs film for 30 s, followed by two rinse–spin steps with methanol and once with toluene. Then 2 mg ml$^{-1}$ PTAA with 10% BCF doping solution in toluene was spin-coated onto InSb/InP CQDs film at 2000 rpm for 30 s. Finally, a Kurt J. Lesker NANO 36 system was used to deposit 10 nm MoO$_x$ as the electronic blocking layer and 100 nm Au as the top electrode with a device area of 3.1 mm$^2$. The smaller device was deposited using the special shadow mask.

**Device characterization.** All the device characterizations were performed in air under ambient conditions. Current-voltage (I-V) measurements were performed with a Keysight Semiconductor Parameter Analyzer (B1500A) with the devices kept in a shield box. The EQE was measured using a Newport Cornerstone 260 monochromator, a Thorlabs MC2000 chopper, a Stanford Research SR570 transimpedance amplifier, and a Stanford Research SR830 lock-in amplifier. Calibrated Newport 818-UV, 818-IR, and S148C photodetectors were used as the reference. For the noise measurements, the frequency-dependent current noise spectrum was measured at low frequencies using the transient-current fast Fourier transform (FFT) method. The measured room-temperature specific detectivity (D*) was calculated according to:

$$D^* = \frac{R\sqrt{A\Delta f}}{i_n}$$

where D* is expressed in units of Jones, R is responsivity, A is the active area of the photodetector, $i_n$ is the noise current spectral density, and $\Delta f$ is the noise bandwidth, here being 1 Hz.

## Data availability
The experimental data that support the findings of this study are available in a public repository https://doi.org/10.34810/data3042.

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

## Acknowledgements

G.K. acknowledges financial support from the European Research Council (ERC) under the European Union's Horizon 2020 research and innovation programme (grant agreement no. 101002306), the European Union under grant agreement No 101119489 (2DNeuralvision), and Project PID2024-161119OB-I00 funded by MICIU/AEI/ 10.13039/501100011033/FEDER, UE. We also acknowledge support from the Fundació Privada Cellex, the program CERCA, and the 'Severo Ochoa' Centre of Excellence. Views and opinions expressed are, however, those of the author(s) only and do not necessarily reflect those of the European Union. Neither the European Union nor the granting authority can be held responsible for them.

## Author contributions

G.K. supervised and directed the study. L.P. and G.K. conceived the idea, designed this study, and co-wrote the manuscript, with feedback from the co-authors. L.P. synthesized the materials, performed the material characterization, analyzed the data, fabricated and characterized the devices. M. D. contributed to the PL and PL lifetime characterizations. D. M. contributed to device characterization. H.W. and A.M. deposited $TiO_2$.

## Competing interests

G.K. serves as co-founder, shareholder, and scientific advisor at Qurv. The remaining authors declare no competing interests.
