## [Transparent Peer Review file · Nature Communications]

Synthesis of monodisperse InSb colloidal quantum dots by monomer concentration control for short-wave infrared photodetectors

Corresponding Author: Professor Gerasimos Konstantatos

Version 0:

Reviewer comments:

Reviewer #1

(Remarks to the Author)
See attached report

Reviewer #2

(Remarks to the Author)

This paper describes a slightly modified synthetic approach for the preparation of InSb QDs of narrow size distribution. More precisely, the work uses the same procedure as that reported by Busatto et al. (Ref. 12) with the difference of modulating the injection speed of the In-Sb mixed precursor. Injecting part of the precursor quickly to induce the nucleation and the rest with a slower rate to keep the monomer concentration high during the growth rate is now relatively widely used for III-V QDs such as InAs.

Even though the novelty of the approach is hence not that great (contrarily to the what is stated in the text), the obtained absorption spectra are better defined than the reported ones with InSb QDs obtained using the $\text{Sb}(\text{NR}_2)_3$ precursor. Concerning the photodetector part, this work is very similar to the paper just published by the same authors in *Advanced Science*, with a slightly longer wavelength (larger particle size) used here. Therefore, all in all, the novelty of this paper appears rather limited, even though some improvement of the InSb QD properties could be achieved.

Further comments:

1) The claim of monodispersity should be underpinned by giving the size distributions for the different sized samples. How the size histograms in Fig. S9 have been obtained? The particles obtained at 250 and even more at 260° exhibit a pronounced deviation from the spherical shape.

2) What are the reaction yields for the different QD sizes?

3) What would be the advantage of the large Bohr exciton radius of InSb, as mentioned in the abstract and introduction? It seems to be rather a drawback because for accessing longer wavelengths one needs to go to large particle sizes (?)

4) Contrarily to what is said in the introduction, the long nucleation period, overlapping with growth, has been well documented for III-V QDs, in particular for InP (cf. works of Cossairt, Owen, etc.)

5) Some expressions sound quite unusual and should be rephrased, like "continuous nucleation growth mechanism" and "long-time-lasting monomers" as well as "more intense ligand adsorption-desorption equilibrium". Also MCC is an unlucky abbreviation for "monomer-concentration-control": it has been used for a long time for metal chalcogenide complex (inorganic ligands for QD surface functionalization).

6) p. 5/6: the authors mention the fusion of QDs as if it were the most natural thing to occur. This discussion should be more

detailed.

7) The discussion uses the so-called Avrami formula for describing the growth kinetics. The approach is not well described neither in the text nor in the methods section. In particular, it is unclear what Figs. 1d) and i) present and how the constant K was chosen in the model.

8) In the caption of Fig. 1, the reaction temperature needs to be specified. The spectra in Fig. 1g do not look as well defined as those presented in Fig. 2a, please comment.

9) In Fig. 2c) one has the impression that the biggest particles are obtained for the lowest temperature.

10) "highly monodisperse" used throughout the text is a pleonasm: monodisperse is already the superlative in terms of size distribution.

11) p. 2/l. 57-59: the explanations given above are valid because reducing the size distribution and avoiding surface oxidation are exactly the two reasons mentioned in the phrase before.

Reviewer #3

(Remarks to the Author)

InSb colloidal quantum dots are very promising materials for optoelectronic applications in the short-wave infrared window, exhibiting wide spectral tunability, solvent tolerance and thermal stability, complementary metal-oxide semiconductor (CMOS) integrability and compliance with RoHS regulations. However, their synthesis still remains underdeveloped. The synthesis protocols currently available typically yield samples with large size polydispersity and limited size tuneability. In this contribution, Konstantatos and coworkers report on a method that yields ensembles with narrow size dispersion emitting in the 950 to 1,900 nm spectral range. Moreover, the authors use these QDs to fabricate SWIR detectors with record efficiency.

The work has been carried out very thoroughly and provides novel and significant insights that make it of high interest to a broad scientific community. However, there are several issues that should be addressed prior to its publication.

I will address my concerns in more detail below.

1. The claim that their work "resolves longstanding questions on InSb CQD growth mechanisms" is exaggerated and unsubstantiated. The evidence presented in the paper to support the notion that the formation of InSb quantum dots follows fully classical pathways is mostly circumstantial and is open to different interpretations. Moreover, the authors should be aware that their claims cannot be generalized because the nucleation and growth mechanisms are extremely dependent on the exact details of the reaction conditions. There have been many recent studies challenging the applicability of the classical nucleation theory to the formation of colloidal nanocrystals of semiconductors (e.g., ref [12], *Nano Letters* 21 (2021) 2487-2496). The authors should thus exercise caution in the way their conclusions regarding the formation mechanism are phrased, making clear that they are valid within the parameter space used in their work. Moreover, they should also be more precise in their discussion: classical nucleation theory concerns "nucleation" not "growth". Therefore, a sentence such as this: "...reveals that the growth mechanism of InSb CQDs follows the classic nucleation theory..." (lines 75-76) is fundamentally incorrect.

2. Exaggerated expressions such as "highly monodisperse" should not be used because they are fundamentally incorrect given that "monodisperse" already implies a single-size distribution. Authors should instead use expressions such "narrow size dispersion" or "small size polydispersity" and support their claims with quantitative information, i.e., provide the actual size dispersion obtained from the size histograms (e.g., "5% polydispersity").

3. The fits to the size histograms presented in Supplementary Figure 9 are mostly poor and therefore do not support the claim of narrow size dispersions. All size histograms are fitted to Gaussians even though many of them are clearly non-gaussian distributions. In other cases, the baseline of the fit is at 10% instead of 0%.

4. It is unclear whether the graphs representing the evolution of the monomer concentration in time (Figures 1e,j, and Figure 2c) are based on experimental data or are simply hypothetical, schematic representations. If they are based on experimental data then the authors should explain how this data was obtained (i.e., how did they measure the "monomer concentration" at different temperatures and times?). If they are just schematics, the authors should explicitly mention it, to avoid misleading readers.

5. The caption of Figure 2j,k is unclear: HRTEM images of which samples?

6. Typo in line 146: "Figure 1b" should be "Figure 2b"

7. Typo in line 172: "Figure 4l" should be "Figure 3l"

8. The authors also resorted to size selective precipitation to obtain the very narrow size dispersions reported in the paper. However, it is unclear whether the absorption spectra shown in the paper and used to support the claim of very narrow size

dispersions were obtained from the crude, unprocessed samples, or from samples subjected to size-selective fractionation. This information is essential to provide support to the formation mechanism proposed by the authors and should have been given.

9. The conclusion regarding Supplementary Figure 8 (lines 151-155) is in fact not supported by the figure. The peak absorption at about 1000 nm is indeed lower for the sample obtained at 260 C with respect to that obtained at 225 C. However, the former consists of larger QDs and therefore its absorption spectrum extends to much longer wavelengths. As a result, its integrated absorption cross-section is in fact larger than that of the sample obtained at 225 C. To be able to support the claim made in lines 148-155 the authors would need to know the concentration of QDs. To obtain that information from the absorption spectra the authors should have determined the correlation between the molar absorption coefficient of the QDs at their lowest energy absorption transition and their size, similar to what has been done in the literature for a large number of different QD compositions.

Version 1:

Reviewer comments:

Reviewer #1

(Remarks to the Author)

The authors have responded in great detail and quantity to the requests by all reviewers. I am satisfied with their answers to my remarks. Thus, I recommend the acceptance of this revised manuscript for publication in Nature Communications.

Reviewer #2

(Remarks to the Author)

The authors carefully responded to all concerns I raised and made the appropriate changes in their manuscript. I recommend the publication of this article.

Reviewer #3

(Remarks to the Author)

The authors have adequately addressed all my concerns. I therefore think that publication of the work in its present form is warranted.

POINT-BY-POINT RESPONSE TO REVIEWERS

Table of Contents

Responses to the comments from Reviewer #1.....	2
Responses to the comments from Reviewer #2.....	10
Responses to the comments from Reviewer #3.....	17

Responses to the comments from Reviewer #1.

Synthesis of monodisperse luminescent colloidal InSb quantum dots by monomer concentration control for short-wave infrared photodetectors by Peng et al. The authors describe a surprisingly simple yet effective adjustment to a previously published synthesis of InSb nanocrystals by them. The quality gain in terms of the optical properties of this underexplored IR-semiconductor with substantial prospects for commercialization in IR photodetectors due to its compatibility with RoHS standards is remarkable. The EQE of their photodetectors is unsurpassed for this material and worthy of a publication in Nature Communications. The paper is compact, straight-to-the-point and mostly sufficiently detailed to ensure reproducibility of the results in other labs.

RE: We sincerely thank the reviewer for your positive feedback and constructive comments, which have greatly enhanced our manuscript quality.

Where this is currently not the case, I have provided suggestions for improvements below. In summary, I recommend publication of the work after the following minor revisions:

1) The biggest weakness is the currently insufficient characterization of the InP-shell employed to the InSb nanocrystals used in the actual photodetectors. The authors mention that the photodetector performance of even their best bare InSb nanocrystals is poor and that shelling with InP is mandatory. Then, it is mandatory to characterize it in a form that lives up to the standards they have demonstrated for the pure InSb nanocrystals. In a revised manuscript, I would like to see for the core-shell particles used in the photodetectors: TEM images, XRD, PL, PL quantum yield, and PL lifetime.

RE: We thank the reviewer for this insightful comment. This new data has been added in Figure S16 and S18.

Supplementary Figure 16. Characterizations of 1360 nm InSb CQDs before and after InP shell growth. TEM images of InSb CQDs (a) and InSb@InP core-shell CQDs (b). (c) STEM image of InSb@InP core-shell CQDs and the corresponding EDX-mapping that shows the atomic ratio of In : Sb : P is 56 : 30 : 14. (d) XRD patterns of InSb CQDs before and after InP shell growth. (e) PL spectra of InSb CQDs before and after InP shell growth, and it shows the PLQY improves from 2% to 6.8% according to the integrating sphere calculation method. (f) The PL lifetime of InSb CQDs before and after InP shell growth. ←

Supplementary Figure 18. Characterizations of 1440 nm InSb CQDs before and after InP shell growth. TEM images of InSb CQDs (a) and InSb@InP core-shell CQDs (b). (c) STEM image of InSb@InP core-shell CQDs and the corresponding EDX-mapping that shows the atomic ratio of In : Sb : P is 54 : 30 : 16. (d) XRD patterns of InSb CQDs before and after InP shell growth. (e) PL spectra of InSb CQDs before and after InP shell growth, and it shows the PLQY improves from 1.5% to 5.2% after shelling. (f) The PL lifetime of InSb CQDs before and after InP shell growth. ←

2) The provenance of the fit in Figure 4b) is a mystery to me. According to the authors, this evolved from applying tight-binding calculations from a 20+ year old paper on lead chalcogenide nanocrystals? This item requires more detail, and the claim “the band gap of InSb CQDs is plotted and perfectly fitted by tight-binding calculation” needs to be substantiated.

RE: We appreciate the insightful comment and sorry for the confusion. This part has been elaborated in the revised version as follows:

“We first attempted to do the fitting by the theoretical model based on an eight-band effective-mass approximation (EMA) approach, proposed by Efros and Rosen,^[29] which simultaneously takes into account the nonparabolicity of the electron- and light-hole dispersion and the coupling between conduction and valence bands. However, the fitting is

unsuccessful and shows significant deviation that is in consistence with the report of Busatto et al.^[12] Subsequently, we turned to the tight-binding calculations model proposed by Allan and Delerue for fitting,^[30] which has been shown to be well suited for InSb CQDs.^[12] The tight-binding calculation is described as follows:

$$E_g(d) = E_g(\infty) + \frac{1}{ad^2+bd+c} \quad (1)$$

where $E_g(d)$ and $E_g(\infty)$ are the band gap (in eV) for a QD of diameter d (in nm) and for bulk, respectively, and a , b , and c are constants. As shown in Fig. S13, Equation 1 has been successfully used to fit the experimentally observed size-dependent trends for InSb CQDs (Equation 2).

$$E_g(d) = 0.17 + \frac{1}{0.0065d^2+0.1615d+0.422} \quad (2)$$

However, as an empirical size dependent equation, the a value of 0.0065 for d^2 is rather small, we therefore – for simplicity- fitted the $E_g(d)$ solely as a function of $1/d$.

As shown in Equation 3 (Fig. 4b), the empirical size-dependent band gap of InSb CQDs can be described as:

$$E_g(d) = 0.17 + \frac{1}{0.2212d+0.3032} \quad (3)''$$

Supplementary Figure 13. Size dependence of the band gap (E_g) of InSb CQDs obtained by MCCA and the fitting with tight-binding calculations by plotting the $E_g(d)$ as a function of $1/d^2$.

Figure 4b. Size dependence of the band gap (E_g) of InSb CQDs obtained by MCCA and the fitting with tight-binding calculations by plotting the $E_g(d)$ as a function of $1/d$.

3) For a group as skilled in photodetector characterization as the authors of this manuscript, I would find it appropriate to provide at least a responsivity or detectivity value for their photodetector.

RE: We thank the reviewer for this insightful comment. This new data have been added as Figure S17 and S19 in the revised version.

Supplementary Figure 17. Device performance characterizations of 1475 nm InSb/InP core-shell CQDs based photodetector. The responsivity spectra (left) of the device with the bias of 0 V and 1 V (reverse bias). Frequency dependent noise spectral density of the device at zero bias (middle), which is measured by transient-current fast Fourier-Transform (FFT) method. The obtained noise spectrum shows a $1/f$ noise dominating at low frequency and reaches a flat noise floor of $\sim 1.4 \times 10^{-13} \text{ A Hz}^{-0.5}$ at a frequency of 1 Hz. Specific detectivity spectrum ($D^* \sim 1.3 \times 10^{11} \text{ Jones}$) of the device (right) at 1 Hz according to the noise spectrum at 0V bias.

Supplementary Figure 19. Device performance characterizations of 1570 nm InSb/InP core-shell CQDs based photodetector. The responsivity spectra (left) of the device with the bias of 0 V and 1 V (reverse bias). Frequency dependent noise spectral density of the device at zero bias (middle), which is measured by transient-current fast Fourier-Transform (FFT) method. The obtained noise spectrum shows a $1/f$ noise dominating at low frequency and reaches a flat noise floor of $\sim 3 \times 10^{-13} \text{ A Hz}^{-0.5}$ at a frequency of 1 Hz. Specific detectivity spectrum ($D^* \sim 3.3 \times 10^{10} \text{ Jones}$) of the device (right) at 1 Hz according to the noise spectrum at 0V bias. ←

4) I do not believe that the discussion of the “peak-to-valley” ratio adds anything to the work. This is a rather unspecific number that depends heavily on the details of the band structure and how it changes with quantum confinement. I am especially critical of their comparison with reference 29, a work on the rather different material cadmium chalcogenides with excitonic absorption in an entirely different part of the spectrum, which ignores almost 25 years of further development of such nanocrystals. The authors have done a good job in characterizing the light-heavy hole splitting and PL FWHM; this is more representative than the peak-to-valley ratio.

RE: We appreciate this insightful suggestion. The comparison with Cd chalcogenides is potentially confusing and have therefore removed it from the revised manuscript. We also fully agree that the observation and discussion of light-heavy hole splitting and PL FWHM are more convincing than referring to the “peak-to-valley” ratio. Accordingly, we have revised Figure 4 to place greater emphasis on these aspects (the revised data are shown in the response to Comment 6).

The “peak-to-valley” ratio was originally included primarily to enable comparison with previously reported InSb CQDs, as many earlier studies did not resolve light–heavy hole splitting or report PL spectra, likely due to limitations in synthetic control, whereas absorption spectra were generally available. Nevertheless, to avoid overemphasis on this metric, we have de-emphasized this discussion and removed the comparison with cadmium chalcogenides in the revised version.

5) I would appreciate a comment on the changing Stokes shift with nanocrystal size.

RE: This is an interesting observation. We have added a comment, as per the referee’s suggestion, which has been mentioned in the revised version as follows: “The Stokes shift behavior of InSb CQDs (i.e. the decreasing Stokes shift with size) may be likely due to a singlet triplet splitting of the exciton states by the electron-hole exchange interaction instead of difference in energy between the exciton states formed with S and P hole states, as reported by Bagga et al for InAs QDs.^[34] However, we also believe that this observation merits further theoretical and experimental investigations to pinpoint the origin of this effect for the case of InSb CQDs in the future.”

6) It would be better if the light-heavy hole splitting values, PL FWHM, and Stokes shifts were presented in a separate table rather than within Figure 4 with poor readability.

RE: We are really sorry for this inconvenience and thank the reviewer for this insightful comment. Figure 4 has been readjusted and a new table (Table S1) has been added as follows:

wavelength (nm)	960	1030	1160	1260	1370	1470	1620	1770
E_g (eV)	1.3	1.2	1.07	0.98	0.90	0.84	0.76	0.7
Size (nm)	2.61	3	3.66	4.18	4.77	5.34	6.22	7.15
$\Delta E_{\text{LH-HH}}$ splitting (meV)	/	/	169	157	137	117	110	96
PL FWHM (meV)	122	115	110	94	96	92	/	/
PL stokes shift (meV)	92	68	56	38	28	18	/	/

Supplementary Table 1. The size-dependent (E_g) light-heavy hole splitting values, PL FWHM, and Stokes shifts.

Fig. 4 | Optical properties of monodisperse InSb CQDs. (a) Absorption spectra of representative InSb CQDs with band gap ranging from 1.3 eV to 0.65 eV. The absorption spectra show the sharpest first absorption peaks among all InSb CQDs. (b) Size dependence of the band gap (E_g) of InSb CQDs obtained by MCCA and the fitting with tight-binding calculations by plotting the E_g (d) as a function of d . (c-d) Absorption spectra showing the valence band splitting energies (HH and LH) as a function of CQD size. (e-f) The absorption and photoluminescence spectra of InSb CQDs with band gap ranging from 1.21 eV to 0.843 eV. The PL is clearly due to band-edge recombination since the Stokes shift is small (20 to 90 meV depending on the size). The full-width at half-maximum (FWHM) of the PL peaks is extremely narrow (100 ± 10 meV) again support the highly monodispersity of InSb CQDs.

Responses to the comments from Reviewer #2.

This paper describes a slightly modified synthetic approach for the preparation of InSb QDs of narrow size distribution. More precisely, the work uses the same procedure as that reported by Busatto et al. (Ref. 12) with the difference of modulating the injection speed of the In-Sb mixed precursor. Injecting part of the precursor quickly to induce the nucleation and the rest with a slower rate to keep the monomer concentration high during the growth rate is now relatively widely used for III-V QDs such as InAs.

Even though the novelty of the approach is hence not that great (contrarily to the what is stated in the text), the obtained absorption spectra are better defined than the reported ones with InSb QDs obtained using the Sb(NR₂)₃ precursor. Concerning the photodetector part, this work is very similar to the paper just published by the same authors in Advanced Science, with a slightly longer wavelength (larger particle size) used here. Therefore, all in all, the novelty of this paper appears rather limited, even though some improvement of the InSb QD properties could be achieved.

RE: We sincerely appreciate the reviewer for the thoughtful comments and the opportunity to clarify the novelty of our manuscript.

Actually, our synthetic procedure is not the same as that reported by Busatto et al. (Ref. 12). They employ a hot-injection method, whereas our work is based on a continuous-injection approach. We just use the same chemicals and precursors as it is well known that there are very limited precursor options for the synthesis of InSb CQDs.

We agree that modulation of precursor injection rates has precedent in the case of InAs CQDs. However, simply applying the InAs method to the synthesis of InSb does not yield ideal results. Specifically, as demonstrated in our previous works (Refs. 15 and 19), even when using continuous-injection strategy, the availability of monodisperse InSb CQDs has

remained restricted to relatively small sizes (corresponding to excitonic peaks below $\sim 1.2 \mu\text{m}$), while larger particles typically exhibit poorly defined absorption features.

To address the existing challenges in the synthesis of InSb CQDs, we believe it is essential to understand the mechanisms that govern CQDs formation and growth. To the best of our knowledge, this is the first study that quantitatively links InSb CQD size dispersity to continuous nucleation via kinetic analysis, and demonstrates that precise control of the monomer concentration can yield sharply resolved excitonic features up to $1.9 \mu\text{m}$, well beyond what was previously reported for InSb CQDs.

Regarding the device section, we agree that it builds upon our recent Advanced Science work as this device architecture works pretty well for InSb CQDs. We have clarified this relationship explicitly and emphasize that the present manuscript extends the operating wavelength beyond $1.5 \mu\text{m}$ and establishes the synthetic improvements as the enabler, rather than introducing a new device architecture.

Further comments:

- 1) The claim of monodispersity should be underpinned by giving the size distributions for the different sized samples. How the size histograms in Fig. S9 have been obtained? The particles obtained at 250° and even more at 260° exhibit a pronounced deviation from the spherical shape.

RE: The size distribution histograms in Fig. S9 (original version) are corresponding to the TEM images presented in Figures 3d-k. The analysis was done by selecting the particles area in TEM images and simulated with the sphere model by Image J software. We apologize for the mistake of fitting in the original version, which resulted from an incorrect x-axis scale and led to deviation from a Gaussian distribution. This data has been modified as Fig. S10 in the revised version as follow:

Supplementary Figure 10. Size distribution histograms of the InSb CQDs corresponding to TEM images in Figure 3. The high quality InSb CQDs obtained by the new method shows small size polydispersity around 10%, the broad size distribution under high precursor concentration condition is due to the fusion of QDs.

2) What are the reaction yields for the different QD sizes?

RE: It has been mentioned in the revised version in Methods section as follows: “The precipitate was collected and redispersed in 2 mL anhydrous toluene with the concentration of 50 mg mL⁻¹, yielding ~100 mg InSb CQDs per synthesis. The reaction yields of different QD sizes are summarized in Table S2.”

wavelength (nm)	1050	1148	1260	1364	1464
Size (nm)	3.1	3.6	4.18	4.76	5.32
amount of precursor for synthesis (mmol)	0.6	0.6	0.6	0.6	0.6
obtained InSb CQDs after purification (mg)	112	104	100	106	89
reaction yield (%)	78.8%	73.2%	70.4%	74.6%	63%

Supplementary Table 2. The reaction yields of different InSb CQDs sizes. The reaction yield is calculated by dividing obtained InSb CQDs after purification (mg) by 0.6 mmol InSb precursors.

3) What would be the advantage of the large Bohr exciton radius of InSb, as mentioned in the abstract and introduction? It seems to be rather a drawback because for accessing longer wavelengths one needs to go to large particle sizes (?)

RE: We thank the reviewer for highlighting this subtle point. What we meant to say was that this is beneficial for the short-wave infrared, not necessarily for longer infrared wavelengths. The combination of the low bandgap and large Bohr radius leads to the fact that InSb QDs of about 3-7 nm in diameter allow coverage of the SWIR spectrum, within a relatively narrow size distribution parameter space. The latter is beneficial for technology development towards device fabrication processes in terms of QD suspensions, film morphology, uniformity, ligand exchange process strategies etc. Moreover, once monodispersity is achieved, this same property enables facile spectral tuneability, allowing targeted design across specific wavelengths of the SWIR spectrum in the quantum-confined regime. This characteristic is particularly advantageous for light emission applications. To clarify the matter we have replaced the word *infrared* with the term *short-wave infrared*, in the abstract.

4) Contrarily to what is said in the introduction, the long nucleation period, overlapping with growth, has been well documented for III-V QDs, in particular for InP (cf. works of Cossairt, Owen, etc.)

RE: We appreciate the reviewer for pointing out this. We carefully checked the literatures and these two papers (Chem. Mater. 2020, 32, 4358-4368; Chem. Mater. 2023, 35, 6152-6160.) are cited properly and introduced in the revised version as follows: “ However, the impact of continuous nucleation on the size and size distribution of II-VI and IV-VI QDs can be readily mitigated either by selecting appropriate precursors^[23] or through homogenization/alloying due to the nature of ionic bond in ionic QDs compounds.^[24-25] In the case of III-V QDs - InP in particular, the continuous nucleation event has also been observed,^[35-36] and efforts to address it have been based on tuning the reactivity of the metal cation precursors.^[36] In contrast, for the InSb CQDs, with the strongest covalent bond among all III-V semiconductors and with limited precursor options, makes the impact of continuous nucleation particularly challenging to address.”

5) Some expressions sound quite unusual and should be rephrased, like "continuous nucleation growth mechanism" and "long-time-lasting monomers" as well as "more intense ligand adsorption-desorption equilibrium". Also MCC is an unlucky abbreviation for "monomer-concentration-control": it has been used for a long time for metal chalcogenide complex (inorganic ligands for QD surface functionalization).

RE: We thank the reviewer for this comment, we have modified the expressions in the revised version. For example: "continuous nucleation growth mechanism" has been replaced by "growth mechanism dominated by continuous nucleation", "long-time-lasting monomers" has been replaced by "persistent monomer supersaturation", "more intense ligand adsorption-desorption equilibrium" has been rephrased as "accelerated ligand exchange dynamics."

To address the confusion about the MCC abbreviation, we have modified it to "monomer-concentration-control approach (MCCA)" in the revised version.

6) p. 5/6: the authors mention the fusion of QDs as if it were the most natural thing to occur. This discussion should be more detailed.

RE: We thank the reviewer for this insightful comment. This part has been elaborated in the revised version as follows: "Furthermore, considering the observation of fusion between QDs due to the accelerated ligand exchange dynamics under high-temperature conditions,^[27-28] we sought to tune the size of InSb CQDs by increasing the amount of precursor at relatively lower temperatures. This motivation arises from the fact that oleylamine is a relatively weak ligand for InSb CQDs, necessitating the addition of oleic acid during purification to prevent CQD aggregation.^[10] Moreover, CQD fusion has been shown to occur during the synthesis, as evidenced by the band-tail absorption features and the corresponding TEM images reported in our previous work.^[19]"

7) The discussion uses the so-called Avrami formula for describing the growth kinetics. The approach is not well described neither in the text nor in the methods section. In particular, it is unclear what Figs. 1d) and i) present and how the constant K was chosen in the model.

RE: We apologize for the lack of clarity. Actually K is not chosen by us, it is from the fitting. We have expanded the Methods section to explicitly describe this fitting in the revised version as follows:

“Avrami Model Fitting. The growth kinetics of coupled nucleation and growth can be described by the Avrami model, which considers the influence of growth on the nucleation process. The general formula is expressed as:

$$f = 1 - \exp(-Kt^n) \quad (4)$$

where K is a constant representing the overall crystallization rate, including both nucleation rate and growth rate, f is solution-to-solid transition volume fraction (here define as the normalized absorption intensity by integrating the absorption spectra) and n is the Avrami exponent.

By applying a logarithmic transformation, Equation 4 can be converted into the following form:

$$\ln(-\ln(1-f)) = \ln K + n \ln t \quad (5)$$

Fig. 1d and i are plotted according to Equation 5, where f is the normalized absorption intensity and t is the time in Fig. 1c and h, and the slope n yields the Avrami exponent.”

8) In the caption of Fig. 1, the reaction temperature needs to be specified. The spectra in Fig. 1g do not look as well defined as those presented in Fig. 2a, please comment.

RE: We are sorry for the confusion. The reaction temperature is 240 °C and it has been specified in the revised version. Fig. 1g shows the absorption spectrum obtained by taking the aliquots of InSb CQDs at 240 °C. They are crude solutions without any purification and size-selection. The absorption spectra with the label of “purification” in Fig. 1g actually

overlaps with the absorption spectra with the label of “1260 nm” (dash line) in Fig. 2a.

9) In Fig. 2c) one has the impression that the biggest particles are obtained for the lowest temperature.

RE: We are sorry for the confusion. It has been corrected in the revised version.

10) "highly monodisperse" used throughout the text is a pleonasm: monodisperse is already the superlative in terms of size distribution.

RE: Following the suggestion of the reviewer, it has been corrected by the use of terms like “monodisperse” or “narrow-size-dispersed” in the revised version.

11) p. 2/l. 57-59: the explanations given above are valid because reducing the size distribution and avoiding surface oxidation are exactly the two reasons mentioned in the phrase before.

RE: We are sorry for this confusion in our expression that stemmed from the editing process and which has been corrected in the revised version as follows: “The weak first excitonic absorption of InSb CQDs is often attributed to (i) strong quantum confinement effects due to the large exciton Bohr radius, where small size variations broaden the absorption, and (ii) surface oxidation due to InSb sensitivity to water and oxygen.^[14,18] Importantly, prior work^[19] has highlighted the importance of having narrow-size-dispersed InSb CQDs with sharp excitonic features for high-performance SWIR photodetectors.”

Responses to the comments from Reviewer #3.

InSb colloidal quantum dots are very promising materials for optoelectronic applications in the short-wave infrared window, exhibiting wide spectral tunability, solvent tolerance and thermal stability, complementary metal-oxide semiconductor (CMOS) integrability and compliance with RoHS regulations. However, their synthesis still remains underdeveloped. The synthesis protocols currently available typically yield samples with large size polydispersity and limited size tuneability. In this contribution, Konstantatos and coworkers report on a method that yields ensembles with narrow size dispersion emitting in the 950 to 1,900 nm spectral range. Moreover, the authors use these QDs to fabricate SWIR detectors with record efficiency. The work has been carried out very thoroughly and provides novel and significant insights that make it of high interest to a broad scientific community.

RE: We sincerely thank the reviewer for your positive feedback and constructive comments, which have greatly enhanced our manuscript quality.

However, there are several issues that should be addressed prior to its publication. I will address my concerns in more detail below.

1. The claim that their work “resolves longstanding questions on InSb CQD growth mechanisms” is exaggerated and unsubstantiated. The evidence presented in the paper to support the notion that the formation of InSb quantum dots follows fully classical pathways is mostly circumstantial and is open to different interpretations. Moreover, the authors should be aware that their claims cannot be generalized because the nucleation and growth mechanisms are extremely dependent on the exact details of the reaction conditions. There have been many recent studies challenging the applicability of the classical nucleation theory to the formation of colloidal nanocrystals of semiconductors (e.g., ref [12], Nano Letters 21 (2021) 2487-2496). The authors show thus exercise caution in the way their

conclusions regarding the formation mechanism are phrased, making clear that they are valid within the parameter space used in their work. Moreover, they should also be more precise in their discussion: classical nucleation theory concerns “nucleation” not “growth”. Therefore, a sentence such as this: “...reveals that the growth mechanism of InSb CQDs follows the classic nucleation theory...” (lines 75-76) is fundamentally incorrect.

RE: We appreciate this concern and have taken action to amend our manuscript to be more technically accurate. In fact, in our original version we were implying our statements for the specific class of synthesis we have employed i.e. the continuous injection approach. The referee is correct though that this was not conveyed clearly. We have therefore taken action and restructured the introduction of our manuscript to make it more explicit and highlight the novelty and impact of our work in a more technically precise manner. We have also corrected the text about the nucleation theory to explain better what we attempted to convey as follows:

“To address the existing challenges in the continuous injection based synthesis of InSb CQDs and enable high-quality InSb CQDs suitable for advanced optoelectronic applications, it is crucial to understand the mechanisms that govern CQD formation and growth. Currently the growth mechanism for InSb CQDs remains unclear and highly dependent on the synthetic protocol. For instance, for the hot injection approach, Busatto et al. propose the aggregative coalescence growth mechanism of InSb CQDs,^[12] Whereas, Wang et al. propose the non-classical growth mechanism of InSb CQDs which follows an initial formation of amorphous intermediate and a subsequent stepwise crystallization process.^[18] Consequently, better understanding of InSb CQD nucleation and growth is required to guide the development of improved synthetic strategies.

In this work, we shed light on the underlying mechanisms of InSb CQD growth based on the continuous injection approach and describe a new synthetic route that overcomes the shortcomings of previous works. Growth kinetics investigation reveals that the synthesis of InSb CQDs

based on the continuous injection follows the classic nucleation theory and the resultant broad size distribution previously^[15] originates from the presence of continuous nucleation events due to the persistent monomer supersaturation.”

And

“By normalizing the absorption intensity (Fig. 1c) and fitting the experimental kinetics with Avrami formula (Fig. 1d, details of fitting described in Methods), the Avrami exponent n is calculated to be 2.04, indicating the growth mechanism of InSb CQDs in previously reported continuous injection synthesis is dominated by continuous nucleation.^[20]”

2. Exaggerated expressions such as “highly monodisperse” should not be used because they are fundamentally incorrect given that “monodisperse” already implies a single-size distribution. Authors should instead use expressions such “narrow size dispersion” or “small size polydispersity” and support their claims with quantitative information, i.e., provide the actual size dispersion obtained from the size histograms (e.g., “5% polydispersity”).

RE: We apologize for the exaggerated expression in the original version. Following the suggestion of the reviewer, it has been corrected to “monodisperse” in the revised version. Also it has been mentioned as follows: “The high quality InSb CQDs obtained by the new method show small size polydispersity around 10%.”

3. The fits to the size histograms presented in Supplementary Figure 9 are mostly poor and therefore do not support the claim of narrow size dispersions. All size histograms are fitted to Gaussians even though many of them are clearly non-gaussian distributions. In other cases, the baseline of the fit is at 10% instead of 0%.

RE: We apologize for the mistake of fitting in the original version, which resulted from an incorrect x-axis scale and led to deviation from a Gaussian distribution. This data has been modified in the revised version as follows:

Supplementary Figure 10. Size distribution histograms of the InSb CQDs corresponding to TEM images in Figure 3. The high quality InSb CQDs obtained by the new method shows small size polydispersity around 10%, the broad size distribution under high precursor concentration condition is due to the fusion of QDs.

4. It is unclear whether the graphs representing the evolution of the monomer concentration in time (Figures 1e,j, and Figure 2c) are based on experimental data or are simply hypothetical, schematic representations. If they are based on experimental data then the authors should explain how this data was obtained (i.e., how did they measure the “monomer concentration” at different temperatures and times?). If they are just schematics, the authors should explicitly mention it, to avoid misleading readers.

RE: We apologize for the confusion that can be misleading. The graphs of Figures 1e,j, and Figure 2c are circumstantial schematic representations according to the results of Figures 1b, g, and Figure 2a respectively. They are not direct and precise experimental data and these have been mentioned in caption of Figure 1 and Figure 2 in the revised version.

5. The caption of Figure 2j,k is unclear: HRTEM images of which samples?

RE: We are sorry for the unclear description. Figure 2j is the high-resolution TEM image of sphere shaped InSb CQD obtained at 250 °C. Figure 2k is the high-resolution TEM image of quasi-tetrapod shaped InSb CQD obtained at 270 °C due to the fusion. These have been clarified in the revised version.

6. Typo in line 146: “Figure 1b” should be “Figure 2b”

RE: We are sorry for the typo. It has been corrected in the revised version.

7. Typo in line 172: “Figure 4l” should be “Figure 3l”

RE: We are sorry for the typo. It has been corrected in the revised version.

8. The authors also resorted to size selective precipitation to obtain the very narrow size dispersions reported in the paper. However, it is unclear whether the absorption spectra shown in the paper and used to support the claim of very narrow size dispersions were obtained from the crude, unprocessed samples, or from samples subjected to size-selective fractionation. This information is essential to provide support to the formation mechanism proposed by the authors and should have been given.

RE: We appreciate for this insightful comment and apologize for the confusion that can be misleading. Actually, the growth kinetics study is carried out by taking the aliquots for absorption measurement (Figure 1b and 1g) without any size-selection process to support our formation mechanism proposal, hence this claim is solid and reliable. This has been highlighted in the revised version as follow: “The growth behavior is then monitored by taking the aliquots without any size-selection process for absorption measurement (Fig. 1g) to understand the growth kinetics.”

Moreover, it is also true that the very narrow size dispersions of InSb CQDs are obtained after size-selection that aims to separate the fused QDs. In Figure 2 and 3, we provided all the absorption spectra before (dash line)

and after (solid line) size selection, and they don't exhibit significant differences apart from the band tail absorption caused by the fusion.

9. The conclusion regarding Supplementary Figure 8 (lines 151-155) is in fact not supported by the figure. The peak absorption at about 1000 nm is indeed lower for the sample obtained at 260 C with respect to that obtained at 225 C. However, the former consists of larger QDs and therefore its absorption spectrum extends to much longer wavelengths. As a result, its integrated absorption cross-section is in fact larger than that of the sample obtained at 225 C. To be able to support the claim made in lines 148-155 the authors would need to know the concentration of QDs. To obtain that information from the absorption spectra the authors should have determined the correlation between the molar absorption coefficient of the QDs at their lowest energy absorption transition and their size, similar to what has been done in the literature for a large number of different QD compositions.

RE: We sincerely thank the reviewer for this insightful comment. We completely agree that it is better to determine the molar absorption coefficient and calculate the corresponding concentrations of QDs to support that claim. Hence, we provided the size-dependent molar absorption coefficients (ϵ_{E1}) of InSb CQDs at the first exciton transition energy (Figure S8) and calculated the concentration of InSb CQDs synthesized at different temperature using the same precursor amount (Figure S9). These data have been attached below, and the corresponding description in the main text has been elaborated as follow: “To test this hypothesis, we first determined the molar absorption coefficient (ϵ_{E1}) of InSb CQDs at the first exciton transition energy that can be fit by the expression $\epsilon_{E1} = 12103d^{1.756}$ (Fig. S8). Noteworthily, the extinction coefficient scales with dot diameter as $\epsilon \propto d^{1.8}$, which deviates from the theoretical prediction ($\epsilon \propto d^3$), yet is comparable to that reported for other III-V InAs CQDs ($\epsilon \propto d^{1.28}$).^[37] This behavior indicates non-volumetric absorption probably due to the extreme quantum confinement, the large

exciton Bohr radius, surface contributions in the electronic structure of InSb CQDs and merits further investigation. The absorption spectra of InSb CQDs synthesized at 230 °C, 240 °C, and 250 °C using the same precursor amount are compared in Fig. S9. Based on Beer-Lambert law calculation, it is found that the InSb CQDs obtained at higher temperature exhibit lower QDs concentration, indicating fewer nuclei formed, which is consistent with our conclusion.”

Supplementary Figure 8. Size-dependent molar absorption coefficients (ϵ_{E1}) of InSb CQDs at the first exciton transition energy. The molar absorption coefficients were determined by digesting the InSb CQDs for inductively coupled plasma (ICP) measurements and calculated following the procedure reported in Reference 38.

wavelength (nm)	1019	1280	1618
Size (nm)	2.95	4.29	6.21
absorbance value (A)	0.381	0.245	0.22
ϵ_{E1} ($M^{-1} cm^{-1}$)	80891	156130	298924
C_{QDs} (μM)	47.1	15.7	7.36

Supplementary Figure 9. The absorption spectra of InSb CQDs (before size-selection) synthesized at 230 °C, 240 °C, and 250 °C using the same precursor amount. Combining the molar absorption coefficients (ϵ_{E1}) obtained in Figure S8 with the Beer-Lambert law, the concentration of InSb CQDs is calculated. As shown in the table (top), the concentration of InSb CQDs decreased from 47.1 μM to 7.36 μM with the increasing of temperature, indicating fewer nuclei formed, which is consistent with our conclusion.

Synthesis of monodisperse luminescent colloidal InSb quantum dots by monomer concentration control for short-wave infrared photodetectors by Peng et al.

The authors describe a surprisingly simple yet effective adjustment to a previously published synthesis of InSb nanocrystals by them. The quality gain in terms of the optical properties of this underexplored IR-semiconductor with substantial prospects for commercialization in IR photodetectors due to its compatibility with RoHS standards is remarkable. The EQE of their photodetectors is unsurpassed for this material and worthy of a publication in Nature Communications. The paper is compact, straight-to-the-point and mostly sufficiently detailed to ensure reproducibility of the results in other labs. Where this is currently not the case, I have provided suggestions for improvements below.

In summary, I recommend publication of the work after the following minor revisions:

- 1) The biggest weakness is the currently insufficient characterization of the InP-shell employed to the InSb nanocrystals used in the actual photodetectors. The authors mention that the photodetector performance of even their best bare InSb nanocrystals is poor and that shelling with InP is mandatory. Then, it is mandatory to characterize it in a form that lives up to the standards they have demonstrated for the pure InSb nanocrystals. In a revised manuscript, I would like to see for the core-shell particles used in the photodetectors:

TEM images, XRD, PL, PL quantum yield, and PL lifetime

- 2) The provenance of the fit in Figure 4b) is a mystery to me. According to the authors, this evolved from applying tight-binding calculations from a 20+year old paper on lead chalcogenide nanocrystals? This item requires more detail, and the claim “the band gap of InSb CQDs is plotted and perfectly fitted by tight-binding calculation” needs to be substantiated
- 3) For a group as skilled in photodetector characterization as the authors of this manuscript, I would find it appropriate to provide at least a responsivity of detectivity value for their photodetector
- 4) I do not believe that the discussion of the “peak-to-valley” ratio adds anything to the work. This is a rather unspecific number that depends heavily on the details of the band structure and how it changes with quantum confinement. I am especially critical of their comparison with reference 29, a work on the rather different material cadmium chalcogenides with excitonic absorption in an entirely different part of the spectrum, which ignores almost 25 years of further development of such nanocrystals. The authors have done a good job in characterizing the light-heavy hole splitting and PL FWHM; this is more representative than the peak-to-valley ratio.
- 5) I would appreciate a comment on the changing Stokes shift with nanocrystal size
- 6) It would be better if the light-heavy hole splitting values, PL FWHM, and Stokes shifts were presented in a separate table rather than within Figure 4 with poor readability.